Temperature vegetation dryness index (TVDI) for drought monitoring in the Guangdong Province from 2000 to 2019

Chen Ailin ailin960518@gmail.com 1 2
Jiang Jiajun 3
Luo Yong 1 2
Zhang Guoqi 4
Hu Bin 1 2
Wang Xiao 5
Zhang Shiqi 6 7
1 Sichuan Earthquake Agency , Chengdu , China
2 Chengdu Institute of Tibetan Plateau Earthquake Research, China Earthquake Administration , Chengdu , China
3 Asia Pacific University of Technology & Innovation , Kuala Lumpur , Malaysia
4 School of Emergency Management, Xihua University, Chengdu, China , Chengdu , China
5 School of Architecture and Civil Engineering, Chengdu University , Chengdu , China
6 College of Earth Sciences, Chengdu University of Technology , Chengdu , China
7 Department of Geosciences and Geography, University of Helsinki , Helsinki , Finland
Wang Jingzhe
Electronic publication date: 2023 Dec 18
Publication date: 2023
Volume: 11
Electronic Location ID: e16337
Received 2023 Jun 6; Accepted 2023 Oct 2
Copyright: ©2023 Chen et al.
Copyright year: 2023
Copyright holder: Chen et al.
License: This is an open access article distributed under the terms of the Creative Commons Attribution License, which permits unrestricted use, distribution, reproduction and adaptation in any medium and for any purpose provided that it is properly attributed. For attribution, the original author(s), title, publication source (PeerJ) and either DOI or URL of the article must be cited.
License URL: https://creativecommons.org/licenses/by/4.0/

Keywords: Drought monitoring, Temperature vegetation dryness index (TVDI), Savitzky–Golay filtering, Remote sensing

Funding: Spark Program of Earthquake Sciences XH23031YC Scientific Research Fund of Institute of Engineering Mechanics, China Earthquake Administration 2020EEEVL0101 This work was supported by Spark Program of Earthquake Sciences (Grant No.XH23031YC), Scientific Research Fund of Institute of Engineering Mechanics, China Earthquake Administration (Grant No. 2020EEEVL0101). The funders had no role in study design, data collection and analysis, decision to publish, or preparation of the manuscript.

==============================
Drought monitoring is crucial for assessing and mitigating the impacts of water scarcity on various sectors and ecosystems. Although traditional drought monitoring relies on soil moisture data, remote sensing technology has have significantly augmented the capabilities for drought monitoring. This study aims to evaluate the accuracy and applicability of two temperature vegetation drought indices (TVDI), TVDINDVI and TVDIEVI, constructed using the Normalized Difference Vegetation Index (NDVI) and the Enhanced Vegetation Index (EVI) vegetation indices for drought monitoring. Using Guangdong Province as a case, enhanced versions of these indices, developed through Savitzky–Golay filtering and terrain correction were employed. Additionally, Pearson correlation analysis and F-tests were utilized to determine the suitability of the Standardized Precipitation Index (SPI) and the Standardized Precipitation Evapotranspiration Index (SPEI) in correlation with TVDINDVI and TVDIEVI. The results show that TVDINDVI had more meteorological stations passing both significance test levels (P < 0.001 and P < 0.05) compared to TVDIEVI, and the average Pearson’R correlation coefficient was slightly higher than that of TVDIEVI, indicating that TVDINDVI responded better to drought in Guangdong Province. Our conclusion reveals that drought-prone regions in Guangdong Province are concentrated in the Leizhou Peninsula in southern Guangdong and the Pearl River Delta in central Guangdong. We also analyzed the phenomenon of winter-spring drought in Guangdong Province over the past 20 years. The area coverage of different drought levels was as follows: mild drought accounted for 42% to 64.6%, moderate drought accounted for 6.96% to 27.92%, and severe drought accounted for 0.002% to 1.84%. In 2003, the winter-spring drought in the entire province was the most severe, with a drought coverage rate of up to 84.2%, while in 2009, the drought area coverage was the lowest, at 49.02%. This study offers valuable insights the applicability of TVDI, and presents a viable methodology for drought monitoring in Guangdong Province, underlining its significance to agriculture, environmental conservation, and socio-economic facets in the region.

Introduction

In the wake of rising global temperatures, there has been a notable uptick in the incidence of extreme meteorological and climatic phenomena, particularly droughts (Asadi Zarch, Sivakumar & Sharma, 2015; Giorgi, Coppola & Raffaele, 2018; Koutroulis, 2019). Droughts cause soil degradation, desertification, water shortages, plant death, dust storms, fires, and other disasters, with serious impacts on agriculture, ecology, the economy, and society (Wilhite, Sivakumar & Pulwarty, 2014; Mishra, Sivakumar & Singh, 2015). Proactive surveillance and assessment of these drought conditions empower governmental entities and associated institutions to devise and implement prompt intervention strategies, bolstering the pillars of consistent socio-economic evolution (Wang et al., 2017; Zhou et al., 2017; Greve et al., 2019).

Traditional drought monitoring mainly relies on soil moisture content data at site scale to characterize the extent and severity of drought in the study area (Mukherjee, Mishra & Trenberth, 2018; Tsiros et al., 2020; Turco et al., 2020). From these observational datasets, an array of meteorological drought metrics have been established, notably the Standardized Precipitation Index (SPI) (Vicente-Serrano, Beguería & López-Moreno, 2010; Dutra et al., 2013), the Standardized Precipitation Evapotranspiration Index (SPEI) (Vicente-Serrano, Beguería & López-Moreno, 2010), and the Palmer Drought Severity Index (PDSI) (Aiguo, Kevin & Taotao, 2004). Nevertheless, the sporadic and imbalanced distribution of meteorological stations impedes the accurate determination of drought’s spatiotemporal dynamics (Palmer, 1965). In recent years, spatial remote sensing technology has made great progress in monitoring drought, with its advantages of rapidly obtaining large-scale spectral information of land cover in time and space, supplementing the shortcomings of traditional monitoring methods (Shiklomanov et al., 2019; Jiao, Wang & McCabe, 2021). It extends the measurement of traditional points to information of surfaces, providing multiple parameters related to the development of drought processes such as precipitation, soil moisture, evapotranspiration, physiological and ecological conditions of vegetation, and surface thermal conditions. Satellite remote sensing monitoring drought technology has made great progress with the rapid development of global earth observation technology. Various remote sensing drought monitoring models and multiple remote sensing drought indices have emerged, which have significant effects in drought monitoring applications in various countries, and are an indispensable means in global drought resistance and disaster reduction (Mishra et al., 2015; Waseem, Ajmal & Kim, 2015; Esfahanian et al., 2017; Alizadeh & Nikoo, 2018). The most widely used one is the Normalized Difference Vegetation Index (NDVI). However, its efficacy diminishes in heterogeneous landscapes due to geographical, ecological, and pedological influences (Liu, Qin & Zhan, 2012; Klisch & Atzberger, 2016). Kogan (1995) introduced the Vegetation Condition Index (VCI) by normalizing the NDVI value to a specific locale’s maximum range. The NDVI component related to weather is smaller than the component related to the ecosystem, so normalization successfully maximizes the reduction of the ecosystem component. VCI can monitor the impact of drought on vegetation health alone, but it is not enough to monitor because it only indicates one humidity condition (Yulistya, Wibowo & Kusratmoko, 2019).

In light of temperature’s potential to partially depict drought scenarios, the Temperature-Vegetation Drought Index (TVDI) has been proposed. This index is formulated on the spectral reflectance properties near-infrared (NIR) and red channels, leveraging the spatial relationship between the land surface temperature (LST) and NDVI as proxies for drought and soil hydration statuses (Tang, Li & Tang, 2010). TVDI, which emerges from the juxtaposition of LST and vegetation index (VI) scatter plots, is exclusively derived from remote sensing datasets, eschewing supplementary data, and exhibits proficiency in regions of moderate or intermittent vegetation. This foundational correlation has proven instrumental in ascertaining soil hydration levels and augmenting drought surveillance endeavors (Sandholt, Rasmussen & Andersen, 2002). TVDI underscores the dynamic interdependence between NDVI and LST. Introducing temperature—a temporally responsive metric—enhances the prediction of hydric stress, potentially modulating NDVI’s representation of vegetation coverage (Bian et al., 2023). TVDI can evaluate soil moisture by synthesizing information from the visible, near-infrared, and thermal infrared spectral bands and combining vegetation indices with surface temperature. Numerous investigations have employed the Temperature Vegetation Dryness Index (TVDI) for a range of applications including fine-scale drought monitoring, crop yield estimation, and soil moisture assessment. Research spanning varied climatic zones—from the arid regions of Northern Africa (Krishnan & Indu, 2023), the Mediterranean climate of Australia (Tao et al., 2021), the temperate zones of Europe (Shi et al., 2020), to the tropical rainforests of South America (Holzman, Rivas & Piccolo, 2014)—underscores the broad adaptability and utility of TVDI in assessing drought conditions. However, the triangle method can be used to monitor drought or estimate soil moisture, the LST-NDVI feature space model is subject to uncertainty due to factors such as terrain changes, which may lead to a lack of clarity in expressing the theoretical margin of the feature space (Sona et al., 2012; Garcia et al., 2014; Holzman et al., 2018). Ran et al. (2005) corrected LST using DEM terrain to construct a feature space and reduce errors caused by terrain issues, improving the applicability of TVDI.

Moreover, the NDVI data used in calculating the TVDI index is affected by soil background in areas with low vegetation coverage and exhibits saturation in areas with high vegetation coverage, and the “maximum value composite method” cannot guarantee the selection of the best pixels, among other shortcomings. The Enhanced Vegetation Index (EVI) based on MODIS data is more sensitive than NDVI in characterizing highly covered vegetation and can effectively resist atmospheric interference, thereby better describing the differences in regional vegetation in different seasons the NDVI input for the TVDI index is susceptible to soil background interference in sparsely vegetated regions and reaches saturation in densely vegetated zones. EVI, derived from MODIS data, surpasses NDVI in discerning areas of dense vegetation and demonstrates enhanced resistance to atmospheric perturbations, thus more accurately delineating seasonal vegetation variations across regions (Tran et al., 2019; Xie & Fan, 2021). Currently, regarding the accuracy and applicability of Temperature vegetation Dryness Index of NDVI (TVDINDV I) and Temperature vegetation Dryness Index of EVI (TVDIEV I), Krishnan & Indu (2023) investigates the utility of TVDI in North Africa. Results show a more promising capability of TVDI for soil moisture (SM) assessment when EVI is used as vegetation index. Yang et al. (2009) discussed the LST-EVI and LST-NDVI feature spaces constructed based on surface temperature and EVI or NDVI and pointed out that TVDIEV I could better characterize soil moisture status in the North China Plain than TVDINDV I. However, Yang et al. (2020) explored the LST-NDVI and LST-EVI feature spaces and inverted soil moisture content in the upper reaches of the Huaihe River, and the results showed that the accuracy of TVDIEV I was lower than that of TVDINDV I. Sun, Chen & Zhang (2014) analyzed the LST-EVI and LST-NDVI feature spaces and compared the drought monitoring results of TVDI based on both with the actual situation in Xilingol League and found that TVDINDV I could better reflect the actual situation in the study area (Sun, Chen & Zhang, 2014). Yang et al. (2020) constructed the LST-NDVI and LST-EVI feature spaces and inverted soil moisture content in the upper reaches of the Huaihe River, and the results showed that the accuracy of TVDIEV I was lower than that of TVDINDV I. A parallel investigation by Sun, Chen & Zhang (2014) into the Xilingol League region supported the findings favoring TVDINDV I. From the current research results, it is not possible to determine the superiority of TVDINDV I and TVDIEV I, so the applicability of drought monitoring using TVDI in specific regions must be discussed.

The Guangdong province, located near the South China Sea, is characterized by a subtropical monsoon climate, manifesting high solar radiation, elevated temperatures, and spatiotemporally heterogeneous precipitation patterns. Despite being one of China’s most precipitation-abundant regions, the province is susceptible to recurrent regional and seasonal droughts, attributable to the imbalanced allocation of rainfall -80% of which is confined to the flood season spanning April to September. This bifurcation into distinct wet and dry seasons renders Guangdong prone to pervasive winter and spring drought events. Robust drought surveillance mechanisms are instrumental in enabling policymakers to delineate the spatiotemporal contours and attributes of drought phenomena, thereby facilitating preemptive and efficacious interventions aimed at mitigating drought-induced detriments and constraining associated economic losses. Although TVDI is currently widely used in drought monitoring studies, there is a lack of selectivity and applicability for TVDINDV I and TVDIEV I in Guangdong Province. This study reconstructs the data for constructing LST-NDVI and LST-EVI feature spaces to ensure the accuracy of the data:usingthe Savitzky–Golay (S_G) weighted filtering technique to reconstruct the NDVI, EVI, and LST data. DEM data used to correct for terrain-induced errors in LST. This study employs the foundational principles of TVDI and integrates the Savitzky–Golay (S_G) weighted filtering technique for refining NDVI, EVI, and Land Surface Temperature (LST) datasets. Upon the foundational principles of TVDI, two vegetation drought indices, TNDINDV I and TVDIEV I were conceptualized and rigorously validated using Standardized Precipitation Index (SPI) and Standardized Precipitation Evapotranspiration Index (SPEI) as meteorological benchmarks. Concomitantly, an exhaustive spatiotemporal analysis of drought patterns in Guangdong Province from 2000 to 2019 was executed. The findings not only offer pivotal insights for the judicious application of TVDI but also furnish invaluable data for policy formulation in areas intersecting drought management and climate change adaptation.

Materials & Methods

This study focuses on drought monitoring in Guangdong Province from 2000 to 2019 using TVDI, which consists of three main steps (Fig. 1). The first step involves data preprocessing and reconstruction: data undergo preprocessing and reconstruction with the utilization of Inverse Distance Weighting (IDW) and Savitzky–Golay (S–G) Filtering, facilitating the synthesis of monthly datasets for NDVI, EVI, and LST. Additionally, a Digital Elevation Model (DEM) is also employed to mitigate terrain effects on LST. The second step involves calculating two types of TVDI, namely TVDINDV I and TVDIEV I, and evaluating them using Pearson correlation coefficients and F-tests based on SPI and SPEI, to select the most suitable vegetation drought index for Guangdong Province. The final phase delves into a comprehensive analysis of the temporal and spatial fluctuations of drought occurrences in Guangdong Province over the 20-year period.

Figure 1 Workflow of this study.

Study area

Guangdong Province, located in southern China (longitude 109°39′∼117°19′East, latitude 20°08′∼25°32′North)(Fig. 2), encompasses temperate, subtropical, and tropical climatic zones. It registers an annual mean temperature of 19–23 °C. Guangdong benefits from extensive solar irradiance, manifesting an with an average daily sunshine duration of 1745.8 h and an annual total solar radiation between 4,200–5,400 MJ/m2, ensuring ample photothermal resources. The province’s topography exhibits higher elevations in the south compared to the north, influencing the spatiotemporal distribution of precipitation. Notably, between April and September, over 80% of the yearly rainfall is recorded. This period witnesses substantial inter-annual variability, where precipitation during pluvial seasons can surpass twice that of arid years. Guangdong is recurrently subjected to climatic adversities such as floods, droughts, and typhoons. Additionally, meteorological events like springtime chill and rainfall, autumnal cold dew winds, and late autumn to early spring frost events and cold surges are prevalent. The province’s drought-afflicted zones vary annually, spanning from a few ten thousand to several hundred thousand hectares, with particularly arid years impacting over a million hectares. By the decade’s close in the 1990s, recurrent droughts affected approximately 24% of the province’s cultivated area, approximating 2.3 million hectares.

Figure 2 Location of the study area.

The boundaries of the study area were sourced from China National Catalog Service For Geographic lnformation (https://www.webmap.cn/commres.do?method=result100W) 1:1 million public version of basic geographic information data (2021) the boundry of the study area; the Digital elevation Model Map of the study area is derived from the Shuttle Radar Topography Mission (SRTM), version V4.1 jointly measured by the United States Space Agency (NASA) and the National Mapping Agency (NIMA) of the Department of Defense. Download from the Chinese Academy of Sciences Geographic Spatial Data Cloud: https://www.gscloud.cn.

Data source and preprocessing

Remote sensing data

The satellite remote sensing data used in this study were derived from the MODIS (Moderate Resolution Imaging Spectroradiometer) sensors onboard the Earth Observing System (EOS) series of satellites launched by NASA (National Aeronautics and Space Administration). The dataset includes surface temperature data (MOD11A2 LST), normalized difference vegetation index (NDVI) (MOD13A3), and enhanced vegetation index (EVI), covering a temporal period from January 2000 to December 2019, or 232 months. The MODIS data products selected encompass data quality control files (QC), indicative of the reliability quality for both LST and NDVI/EVI products. Table 1 shows the data information. Statistical assessments of the LST_QC and NDVI_QC data demonstrate that the mean annual reliability for data quality in Guangdong Province from 2000–2019 exceeds 90%, indicating robust data integrity within the study locale.

Table 1 Data information.

	Datasets	Temporal resolution	Spatial resolution	
LST	MOD11A2	8 days	1 km	
NDVI	MOD13A3	1 month	1 km	
EVI	MOD13A3	1 month	1 km	
DEM	SRTM-DEM		90 m	

The acquired remote sensing data underwent the following preprocessing steps:

(1) Band Extraction: From the procured HDF-formatted MO13A3 and MOD11A2 products, MO13A3 encompasses 11 sub-datasets, including QC files, various band reflectance datasets, and the NDVI/EVI vegetation index. Conversely, MOD11A2 consists of 12 sub-datasets, encapsulating diurnal and nocturnal LST data, along with QC files. For MOD13A3, the NDVI, EVI dataset and pixel reliability QC files were extracted, while for MOD11A2, the daytime LST_Day dataset and corresponding QC_Day files were retrieved for ensuing processing.

(2) Quality Control: Utilizing the pixel reliability from the MOD13A3 QC file, a mask file for the vegetation index was formulated, preserving only high-caliber data with a pixel reliability score of 0. Subsequently, the initial NDVI and EVI data were masked to extract a high-quality vegetation index dataset. For MOD11A2, a mask file was constructed based on the QC_Day file, extracting rasters with binary bits 6–7 set at 00 (signifying an LST average error ≤ 1K).

(3) Affine Transformation: Metadata from the MODIS products provided latitude and longitude coordinates for image vertices, alongside image-generated time information. An affine transformation matrix was constructed to transition from image to geographic coordinates. Each image then underwent this transformation, adopting the WGS 84 coordinate system, resulting in time-stamped tif format data.

(4) Clipping and Resampling: The boundary vector data of Guangdong Province was conformed to the WGS 84 coordinate system of the preprocessed image. The GDAL library’s Warp function was then employed to trim and generate provincial NDVI/EVI/LST data at 1 km resolution. Additionally, the 90m resolution DEM data was adapted and resampled to 1km.

(5) Monthly LST Data Synthesis: Given that MOD11A2 supplies 8-day temperature products in contrast to the monthly NDVI/EVI products, a Python tool utilizing the arcpy.sa library’s CellStatistics function was devised to translate LST data to a monthly metric.

DEM data

The Digital Elevation Model (DEM) utilized in this study were sourced from the SRTM-DEM data product of the Chinese Academy of Sciences Geographic Spatial Data Cloud (http://www.gscloud.cn). Originally at a spatial resolution of 90 m, the data underwent preprocessing—including outlier removal, image mosaicking, and regional clipping—before being resampled to 1km for LST data’s terrain correction (Table 1).

Meteorological data

Monthly average temperature (°C) and precipitation (mm) data were obtained from 25 meteorological stations within the study area, provided by the China Meteorological Administration Meteorological Data Center (http://data.cma.cn). Notably, the Lianping County station was omitted from the study because of substantial data voids. This meteorological data aided in computing SPI/SPEI across varying durations from 1960 to 2019 in Guangdong Province.

The traditional TVDI model

The Temperature Vegetation Dryness Index (TVDI) serves as a pivotal tool in remote sensing drought monitoring. It appraises the conditions of soil surface moisture conditions based upon the spatial features of a two-dimensional scatter plot similar to a triangle, created from the surface temperature (LST) and vegetation index (VI) data. This LST-VI feature space is a scatter plot of LST and VI values for a given pixel in a two-dimensional coordinate system (Fig. 3).

Figure 3 A conceptual land surface temperature (LST)-normalized difference vegetation index (NDVI) triangle (Sandholt, Rasmussen & Andersen, 2002).

Figure 3 illustrates the triangle’s vertices represent three extreme situations in the LST-VI feature space: the upper left corresponds to to arid, barren soil (low VI values and high LST values), the lower left represents wet soil (minimum VI and LST values in that region), and the bottom right vertex symbolizes the most saturated soil with highest vegetation cover (high VI values and low LST values). For bare soil areas, changes in surface temperature are highly correlated with soil moisture changes. From the top left corner to the bottom left corner of the feature triangle, the amount of surface soil water evaporation gradually increases from zero to the maximum value. The trajectory from the upper left to the lower left of this triangle indicates an incremental ascent in surface soil water evaporation. The bottom right vertex represents the surface soil being completely covered by vegetation and having sufficient water content, with a water stress index of 0 in this region (Carlson, Gillies & Perry, 1994). The triangle’s oblique side signifies a compromised soil moisture efficiency, and minuscule surface water evaporation, thereby being characterized as a “dry edge” state during drought periods. Conversely, the base suggests ample soil moisture content, unrestrictive to plant growth, with the surface evapotranspiration being equated to potential evapotranspiration, thus falling within a “wet edge” state (Price, 1990).

By simplifying the feature space, Sandholt constructed the Temperature Vegetation Dryness Index (TVDI), the mathematical expression is shown below: (1) TVDI=Ts−TsminTsmax−Tsmin

Eq. (1), Ts represents the surface temperature value (°C) of any pixel on the remote sensing image, Tsmin represents the minimum surface temperature value of all grid values under the same vegetation cover condition (with a constant VI value), and Tsmax represents the corresponding maximum surface temperature value under the same VI value.

From the remote sensing data amalgamating LST and VI data, the maximum and minimum surface temperature values for the same vegetation index are extracted, and the vegetation indices are linearly fitted to obtain the dry edge (Eq. 2) and wet edge (Eq. 3) equations in the feature space: (2) Tsmin=a1+b1×VI

(3) Tsmax=a2+b2×VI

where, a1 and b1 are the coefficients of the wet edge equation, while a2 and b2 are the coefficients of the dry edge equation.

According to the principle of TVDI, the dry edge corresponds to a TVDI value of 1, while the wet edge corresponds to a TVDI value of 0, and the TVDI value of any point is calculated to be between 0 and 1. The higher the TVDI value, the more severe the corresponding soil drought; the lower the TVDI value, the lighter the corresponding soil drought. The TVDI drought level is classified as wet (0∼0.2), normal (0.2∼0.4), mild drought (0.4∼0.6), moderate drought (0.6∼0.8), and severe drought (0.8∼1.0).

Data reconstruction to the TVDI model

The NDVI, EVI, and LST datasets, once processed for quality control, exhibit missing values in raster cells deemed to have unreliable quality. To build a consistent time series, it is imperative to optimize and smooth these continuous raster values over temporal dimensions. In this study, the Inverse Distance Weight (IDW) interpolation technique is employed to estimate values for the missing raster cells, facilitating spatial data reconstruction. Subsequently, the Savitzky–Golay (S–G) time series filtering approach is applied to refine the temporal data series, ensuring comprehensive time reconstruction of the datasets.

Inverse distance weighted (IDW)

The inverse distance weighting (IDW) method, introduced in the late 1960s, predicated upon the primary principle of geography and was initially conceptualized to forecast regional precipitation patterns. Its application has since expanded, finding utility across diverse geographic spatial analyses. Compared to algorithms that interpolate solely based on the values of proximate data points, IDW considers the values from more distant locations. and assigns weights It allocates weights that are inversely proportional to the distance between the observation and the predicted sites (Jaya et al., 2021).The mathematical formula is as follows: (4) Z= ∑i=1n1DipZi/∑i=1n1Dip

where Z is the predicted value at the point (X, Y), Zi is the actual value of the i (i =1, 2, 3, …, n) point data with coordinates (Xi, Yi), Di is the distance between the prediction point and the i data point (Eq. 5), and p is the power of the distance. (5) Di=X−Xi2+Y−Yi2

Savitzky–Golay (S–G) filtering

The Savitzky–Golay (S–G) filtering method, commonly referred to as the adaptive filter, was proposed based on smoothing time series data and the least-squares principle through convolution computation (Savitzky & Golay, 1964). Unlike conventional sliding window averaging approaches that use constant weighting coefficients, the S-G method determines these coefficients based on the least-squares fit of a specified high-order polynomial within the filtering window,, expressed as Eq. (6): (6) Yj∗=∑i=−mmCi×Yj+iN

where, Yj∗ represents the fitted values of the data set, Y represents the original values of the data set, Ci represents the coefficients for filtering the i data set value, j is the index of the original value, m is half the width of the filter window, and N is the length of the filter, which is equal to the width of the sliding array. (7) N=2m+1

The index of fitting effectiveness is considered the best filtering effect when it reaches its minimum value during the filtering process. The calculation of the index of fitting effectiveness (Eq. 8) after k iterations is as follows: (8) Fk= ∑i=1N|Yik−Yi0|

where, Yi0 is the dataset values without iteration, and Yik is the values in the sequence after the k iteration.

DEM correction to the TVDI model

The accuracy of TVDI in drought assessment is intrinsically related to LSTand V). However, it is also influenced by topographical elevation and geographical latitude, two paramount factors that dictate atmospheric background variance and solar radiation when acquiring LST data via remote sensing instrumentation. Such influences can precipitate direct discrepancies in TVDI computation. Consequently, rectifying LST data is essential to enhance the computational accuracy of TVDI. The formula for correcting LST data is given in (Eq. 9): (9) Tc=Ts+a×H+b×L+c

where, Ts and Tc represent the temperature values before and after correction, respectively. H is the elevation, L is the latitude, a is the elevation correction coefficient, and b and c are the latitude correction coefficients. Figure 4 shows the DEM corrected LST data comparison.

Figure 4 Comparison after terrain correction.

The LST data is derived from the NASA EOSDIS Land Processes Distributed Active Archive Center (https://doi.org/10.5067/MODIS/MOD11A2.006).

Assessment method

This study validates the accuracy of the remote sensing-based TVDI by comparing it with the widely used meteorological drought indices, namely Standardized Precipitation Index (SPI) and Standardized Precipitation Evapotranspiration Index (SPEI). McKee, Doesken & Kleist (2012) leveraged historical records to delineate the probabilistic distribution of cumulative precipitation across temporal intervals, introducing the SPI. This innovation addressed the challenge of juxtaposing standalone rainfall data across disparate spatial–temporal scales and subsequently garnered global acclaim for its efficacy in drought surveillance. Augmenting the SPI framework, Vicente-Serrano, Beguería & López-Moreno (2010) integrated both precipitation and evapotranspiration metrics, heralding the SPEI, a tool proficient in appraising drought scenarios across diverse intervals. Its universal application in global drought research and praxis is attested by subsequent works (Noorisameleh et al., 2020; Shi, Wu & Ding, 2020). Given the multifaceted temporal scales of SPI and SPEI (encompassing 1-month, 3-month, 6-month, 12-month, and beyond), our study has elected January as the representative temporal metric, harmonizing it with the monthly granularity of TVDI.

For accuracy assessment, this study employed the Pearson Correlation Coefficient and F-test. The Pearson Correlation Coefficient, renowned for gauging the linearity between paired variables (X and Y), with values ranging between -1 and 1. Complementing this, the F-test scrutinizes the hypothesis suggesting that a given statistical figure adheres to an F distribution under the null premise (H0). Historically, the Pearson Correlation Coefficient traces its roots to Karl Pearson’s adaptation of a concept posited by Francis Galton in the late 19th century. Concurrently, the F-test, occasionally referenced as the variance ratio test, assesses variance uniformity, producing results consistent with the F-distribution under the null premise (H0).

The calculation formula for the Pearson correlation and F-test are Eqs. (10) and (11): (10) r=∑i=1nXi−X¯Yi−Y¯∑i=1nXi−X¯2∑i=1nYi−Y¯2

(11) F=1n−1∑i=1nXi−X¯2/1n−1∑i=1nYi−Y¯2

Results

TVDINDV I and TVDIEV I construction results

LST-VI feature space construction

In this study, IDW and Savitzky–Golay (S–G) Filtering were applied to reconstruct NDVI, EVI and LST data with the parameters set as shown in the table below, while the parameters for topography correction are shown in Table 2 below:

Table 2 Data reconstruction parameter setting.

	Parameter	Setting value	
IDW	Distance parameters	2	
Variable retrieval radius value	12	
S-G	Half-width of filter window m	7	
Polynomial fitting order d	2	
Terrain correction	Elevation correction coefficient a	0.006	
Latitude correction factor b	0.4	
Latitude correction factor c	−16	

In previous studies, the range of 0.2 < NDVI < 0.8 was directly selected to fit the dry and wet edges of the characteristic space when calculating TVDI values, because NDVI is more sensitive to changes in soil background, and when NDVI < 0, the soil surface layer is mainly water or snow, and surface moisture can be considered 100%. When vegetation cover is less than 20%, NDVI values are difficult to indicate the degree of vegetation cover in the area. When vegetation cover is greater than 80%, the increase in NDVI value will be delayed and reach saturation, resulting in reduced sensitivity to vegetation cover recognition (Sha et al., 2011). However, for Guangdong Province, which has a large population density and wide urban coverage, some extreme cells (such as urban green belts, etc.) will appear in the range of 0.2∼0.8, and these very few cells need to be eliminated in order to make the dry and wet edge fitting equation more reasonable. Moreover, the principle of enhanced vegetation index EVI and NDVI indicating vegetation coverage is different, and the interval fitting of 0.2∼0.8 cannot be directly selected, and the fitting range needs to be selected according to the actual situation of the spatial distribution of the features. In addition, the results show that the pixel histogram (the frequency distribution of pixels with different VI values) is closely related to the shape of the feature space, and plays an important auxiliary role in the determination of dry and wet edges in the feature space (Zhang & Feng, 2012).

Based on this, by extracting the maximum and minimum land surface temperature values corresponding to different NDVI/EVI pixel grid values, the distribution of LST-VI feature space in Guangdong Province from September 2000 to December 2019 is obtained on a monthly basis. Twelve months of feature space and corresponding VI pixel histograms are selected for display, representing the distribution of feature space in different years for each of the 12 months (Figs. 5 and 6).

Figure 5 LST-NDVI monthly feature space and NDVI histogram.

Figure 6 LST-EVI monthly feature space and EVI histogram.

Figure 5 delineates the LST-NDVI feature space, where the x-axis represents NDVI values, and the y-axis represents LST (°C). The shape of the feature space, demarcated by the dry and wet edges, can be characterized as a tapered spindle with acuminated termini. Transitions along the dry and wet edges can be segmented into three distinct phases. Initially, the LST value of the points on the dry edge gradually increases from NDVI = 0, conversely, those on the wet edge descend. Subsequently, with escalating NDVI values, the data points on both edges expand laterally. In the third stage, the points on the dry edge decrease with the increase of NDVI values, while those on the wet edge decrease correspondingly, and finally converge at the maximum NDVI value. Considering the corresponding histogram, the points in the first stage have extremely few pixels and can be ignored when fitting the dry and wet edges. The points in the second and third stages should be reasonably removed according to the distribution of the histogram. Between January and March, the NDVI histogram exhibits a broad distribution, predominantly spanning 0.35 to 0.75, indicative of sparse vegetative cover. Progressing through the months, NDVI value ranges migrate rightward, peaking beyond 0.9, thus extending the LST-NDVI feature space into higher NDVI sectors. By August, the NDVI histogram primarily encompasses a 0.65–0.85 range, manifesting a more pronounced and integral triangle in the feature space. From September, the histogram range shifts to the left, demonstrating significant seasonal characteristics.

Figure 6 illustrates the LST-NDVI feature space, the LST-EVI feature space also exhibits a spindle shape, but the shape changes significantly with the month. From May to October, it is a more elongated spindle shape, while from November to April, it is a shorter and rounder spindle shape. The range of the feature space fluctuates more than that of LST-NDVI, with the maximum value in January and February not exceeding 0.5, reaching almost 0.8 in July and August, and then shifting to around 0.6 in November and December. Combining with the histogram, in the warmer climate from May to October, the EVI values are concentrated between 0.45 and 0.7, while from November to April, they are distributed between 0.35 and 0.6.

Pixels with exceptionally high or low NDVI/EVI values are typically sparse, representing unique soil moisture conditions, making it difficult to ensure that there are different corresponding values from dry to wet at that NDVI. Therefore, the values of the pixels at both ends are extremely unreliable and cannot be used for fitting the dry edge and wet edge (Sha et al., 2011; Zhang & Feng, 2012). When fitting the dry and wet edges, the VI interval with more pixels and after the critical point in the feature space should be selected for fitting. The analysis of the feature spaces constructed by the two vegetation indices shows that there are obvious critical points, such as the NDVI = 0.55 in Fig. 5C, where the values on the dry edge begin to decrease gradually and the values on the wet edge increase gradually. The pixel volume corresponding to NDVI = 0.55 in the histogram also stands robust, making edge fitting from this value onward align seamlessly with TVDI principles.

LST-VI dry and wet edge fitting

In Guangdong Province, the LST-VI feature space exhibits pronounced monthly variations and distinct segmentation. Therefore, it is necessary to adjust the fitting of the dry and wet edges based on the results analyzed in the previous section. During the interval spanning September 2000 to December 2019, the dry and wet edges were reformulated by excluding anomalous NDVI and LST values. For this refinement, the range 0.55 < NDVI < 0.9 was employed within the LST-NDVI feature space. The least squares regression technique was utilized to fit the dry and wet edges, with select outcomes presented in Figs. 7 and 8.

Figure 7 LST-NDVI dry and wet edge fitting.

Figure 8 LST-EVI dry and wet edge fitting.

Figure 9 Correlation of TVDI and SPI-1/SPEI-1 at Dongyuan weather station.

Figurs 7 and 8 illustrate that by setting the fitting range according to the specific characteristics of the feature space in different months in Guangdong Province and removing the points with fewer pixels at both ends of the histogram, the fitting of the dry-wet edge equation was optimized to ensure that there are continuous pixel values from dry to wet for each VI value. The R2 coefficient for the wet edge fitting across various intervals ranges from 0.5 to 0.9, with the mean R2 coefficient for the dry edge being 0.6. This results in a characteristic triangular configuration marked by an inverse relationship in the dry edge and a direct relationship in the wet edge for the dry-wet edge fitting curve. This restructured dry-wet edge feature space offers a notable improvement over the initial space delineated in the earlier segment.

TVDINDV I and TVDIEV I selection

The SPI and SPEI values were computed using monthly precipitation data from 22 meteorological stations in Guangdong Province from 2000 to 2019. The 1-month SPI values were calculated, and the average of the 22 station values was used to represent the province-level 1-month SPI situation. The TVDINDV I and TVDIEV I values of the corresponding pixels for the 22 meteorological stations from 2000 to 2019 were extracted. Correlation analysis and F-tests were then conducted by comparing these values with the SPI and SPEI values for each meteorological station.

Taking the Dongyuan meteorological station as an example, the R-value in Fig. 9 represents the correlation coefficient, and P represents the P-value of the F-test. According to the calculation principle of TVDI, the higher the value, the more significant the drought. In contrast, the meteorological drought index SPI/SPEI is the opposite, with lower values indicating more severe drought. Therefore, the two should show a negative correlation. The Pearson correlation coefficients between the monthly N-TVDI and SPI-1 from September 2000 to December 2019 at the Dongyuan meteorological station were −0.6025, and −0.231 with SPEI-1, both passing the significant test of p < 0.001. To compare the strength of the correlation between N-TVDI/E-TVDI and meteorological drought indices, the correlation coefficients and P-values of TVDI and SPI-1/SPEI-1 at 22 meteorological stations were plotted in a table (Table 3).

Table 3 Correlation coefficients of TVDINDV I and two meteorological indices.

Station	SPI/TVDINDV I	SPEI/TVDINDV I	SPI/TVDIEV I	SPEI/TVDIEV I	
\	Pearson’R	P	Pearson’R	P	Pearson’R	P	Pearson’R	P	
Dongyuan	−0.625	0	−0.231	<0.001	−0.6014	0	−0.2022	0.00196	
Fogang	−0.46649	<0.001	−0.34198	<0.001	−0.44667	<0.001	−0.2924	<0.001	
Gaoyao	−0.07435	0.25937	−0.13074	0.04669	−0.18309	0.00515	−0.0761	0.24855	
Guangning	−0.42331	<0.001	−0.18705	0.00425	−0.47096	<0.001	−0.166	0.01134	
Guangzhou	−0.50902	<0.001	−0.40043	<0.001	−0.5454	0	−0.3943	<0.001	
Huilai	−0.48116	<0.001	−0.21736	<0.001	−0.47721	<0.001	−0.1732	0.00848	
Huiyang	0.0898	0.17285	−0.10194	0.12152	0.15927	0.01517	−0.0175	0.79092	
Lianxian	−0.04324	0.51226	−0.20471	0.00172	−0.25841	<0.001	−0.157	0.01672	
Luoding	−0.49438	<0.001	0.00353	0.95739	−0.18436	0.00485	−0.0082	0.90088	
Meixian	−0.29542	<0.001	−0.16232	0.01331	−0.27879	<0.001	−0.1123	0.00833	
Nanxiong	−0.06684	0.31074	−0.1039	0.11452	0.06944	0.29223	−0.0031	0.96297	
Shantou	−0.23481	<0.001	−0.21967	<0.001	−0.2349	<0.001	−0.1829	0.00521	
Shanwei	−0.6598	0	−0.45609	<0.001	−0.68833	0	−0.436	<0.001	
Shaoguan	−0.12803	0.1	−0.03847	0.55993	−0.1184	0.07192	0.03471	0.59889	
Shenzhen	−0.51947	0	−0.31149	<0.001	−0.5149	0	−0.2646	<0.001	
Taishan	−0.41988	<0.001	−0.24645	<0.001	−0.4311	<0.001	−0.228	<0.001	
Wuhua	−0.47686	<0.001	−0.1371	0.03691	−0.4288	<0.001	−0.0748	0.25662	
Xinyi	−0.3903	<0.001	−0.14663	0.02552	−0.4717	<0.001	−0.153	0.01977	
Xuwen	0.32147	<0.001	−0.01254	0.85003	0.28801	<0.001	−0.0724	0.27394	
Yangjiang	−0.277	<0.001	−0.26654	<0.001	−0.278	<0.001	−0.2665	<0.001	
Zengcheng	−0.2041	0.00194	−0.34243	<0.001	−0.133	0.043	−0.3099	<0.001	
Zhanjiang	−0.43314	<0.001	−0.26937	<0.001	−0.4331	<0.001	−0.2779	<0.001	

Table 3 indicates that among the 22 meteorological stations, TVDINDV I was negatively correlated with SPI in 20 stations with exceptions being Huiyang and Xuwen.The Pearson correlation coefficient for the association between TVDINDV I and SPI in Lianxian did not pass the significance test of P < 0.05. Furthermore,16 stations passed the significance test of P < 0.001. The Pearson correlation coefficient ranged from −0.23 to −0.66, with an average of −0.44. TVDINDV I was negatively correlated with SPEI in 21 stations (excluding Luoding). Four stations (Huiyang, Nanxiong, Shaoguan, and Xuwen) did not pass the significance test of P < 0.05, while coefficients for the remaining stations varied between −0.13 and −0.45.

In the correlation analysis between SPI-1/SPEI-1 and TVDIEV I, 19 stations were negatively correlated with SPI (except for Huiyang, Xuwen, and Nanxiong), and the Shaoguan station did not pass the significance test of P < 0.05. A total of fifteen stations passed the significance test of P < 0.001, with the correlation coefficient ranging from −0.23 to −0.60 and an average of −0.43. TVDIEV I was negatively correlated with SPEI in 21 stations (excluding Shaoguan), and six stations (Gaoyao, Huiyang, Luoding, Nanxiong, Wuhua, and Xuwen) did not pass the P < 0.05 significance test. The correlation coefficient for the remaining stations ranged from −0.11 to −0.43.

Upon evaluating the correlation between the two monthly TVDI drought indices and the corresponding SPI-1/SPEI-1 at each station, it was found that TVDINDV I surpassed TVDIEV I in both the number of stations achieving statistical significance at the P < 0.001 and P < 0.05 levels, and in the average Pearson correlation coefficient. This suggests that TVDINDV I offers a more sensitive response to drought in Guangdong province.

Spatial and temporal distribution characteristics of drought in Guangdong Province from 2000 to 2019

Figures 10 and 11 shows the average spatial distribution pattern of TVDI in Guangdong Province from 2000 to 2019. Predominantly, the province manifests a humid climate. However, a marked contrast in drought severity is evident between its northern and southern regions. The most pronounced drought conditions pervade the southernmost and mid-eastern coastal zones, exhibiting a multifaceted range from mild to severe degrees of drought. The northeastern coast sporadically experiences moderate to acute drought conditions, whereas the northwest to central-west areas remain largely devoid of drought. The Leizhou Peninsula, is the most severely affected area, with Suixi County and Xuwen County experiencing the most severe drought, and moderate to severe drought areas widely distributed. The next most affected area is the southern coastal area of Maoming City, where most of the areas are affected by moderate drought and some areas by severe drought. The drought in the central and eastern areas is mainly concentrated in Dongguan, Shenzhen, Huizhou, and Guangzhou, with the most extreme drought levels occurring in the urban areas of Dongguan, the entire county of Huiyang, the urban areas of Huizhou, and the eastern part of Bao’an County. In addition to the areas with large-scale drought distribution mentioned above, small-scale moderate to severe drought occurs in Huizhou County, Haifeng County, Lufeng County, Chaoyang County, Shantou City, Chenghai County in the northeastern coastal areas, and Fengshun County, Dapu County, and Chaozhou City in the northeastern central area. Apart from these focal areas of extensive drought, localized moderate to severe drought episodes also transpire within the northeastern coastal territories and northeastern central districts.

Figure 10 Multi-year average spatial distribution pattern of TVDI in Guangdong Province from 2000 to 2019.

Figure 11 Distribution of annual average TVDI in Guangdong Province, 2000–2019.

Guangdong Province frequently experiences winter-spring droughts, a phenomenon attributed to diminished rainfall, augmented evaporation, reduced soil moisture, and elevated temperatures during these seasons. To analyze the specific areas affected by these droughts, mean TVDI To analyze the specific areas affected by these droughts, the mean TVDI values from December of the previous year to April of the following year were calculated. The resulting maps depict the distribution of droughts during the winter and spring seasons for each year between 2001 and 2019 (Fig. 12).

Figure 12 TVDI distribution of winter-spring drought in Guangdong Province, 2000–2019.

As shown in Fig. 12, the drought-prone areas during winter-spring transition predominantly align with the annual TVDI distribution, especially within Suixi County of the Leizhou Peninsula, which consistently endures such drought conditions. The spatial distribution and extent of drought vary over time, with the largest affected area occurring in 2003 when the entire southern Leizhou Peninsula was in a severe drought state. The drought eased after 2004 but intensified again in 2005–2006, before subsiding in 2007. In 2009, the winter and spring drought was the lightest, but in 2010, it was the second most severe after 2003, with Xuchang County in the southernmost part of the Leizhou Peninsula shifting from moderate drought in the previous years to severe drought. The area of moderate drought in the northeastern coastal region also significantly expanded. The drought gradually eased until 2013, but intensified again in 2014, with severe impacts continuing until 2019, with winter and spring droughts affecting large areas of Guangdong Province almost every year. For a quantitative delineation of the extent and intensity of winter-spring droughts in Guangdong Province, an annual assessment of the impacted area’s proportion, stratified by drought severity, was executed (Fig. 13). The proportion of areas unaffected by winter and spring droughts in Guangdong Province ranged from 15.78% (2003) to 50.98% (2009), with a significant fluctuation range. The proportion of areas affected by mild drought ranged from 42% to 64.6%, with moderate drought ranging from 6.96% to 27.92%, and severe drought ranging from 0.002% to 1.84%. In 2003, the drought affected the largest area, accounting for 84.2% of the province’s total area, followed by 2010, with the drought affecting 74.98% of the province’s area. The lowest coverage occurred in 2009, with only 49.02% of the province’s area affected by drought. Although the proportion of severe drought areas in Guangdong Province is not high, the Leizhou Peninsula region experiences severe droughts every winter and spring, causing significant damage to local agriculture.

Figure 13 Area share of the degree of drought in winter and spring consecutive droughts in Guangdong Province, 2000–2019.

Discussion

Drought manifestations possess unique spatial and temporal characteristics. However, different drought monitoring indices have obtained different results. Currently, researchers often use the TVDI to monitor drought conditions and combine it with vegetation indices such as NDVI or EVI and other vegetation indices for integrated analysis. Despite its widespread use, there remains a divergence of opinion regarding the suitability of NDVI and EVI in relation to TVDI. NDVI’s accuracy is perceived to be constrained in densely urbanized or desertified areas experiencing acute drought. Conversely, EVI has been shown to effectively negate the influences of atmospheric scattering and soil background, with minimal interference from non-vegetative cover. Therefore, it is inferred that EVI may be relatively more suitable in Guangdong Province, while there are studies using TVDIEV I for drought monitoring in Guangdong Province, but there is no in-depth discussion on who is more suitable for the two drought indices (Wang et al., 2014), TVDINDV I and TVDIEV I. However, in our study, we analyzed the correlation results between the two drought indices of TVDINDV I and TVDIEV I and the corresponding station monthly scale SPI-1/SPEI-1, and we found that TVDINDV I has more stations than TVDIEV I on both significance test levels P < 0.001 and P < 0.05, and the correlation coefficient Pearson ’s mean value of R is also slightly higher than that of TVDIEV I. It indicates that TVDINDV I responds better than TVDIEV I to drought in Guangdong Province, so for different data scenarios, it may be necessary to select the appropriate vegetation index for drought monitoring studies in conjunction with specific situations.

The TVDI serves as a prominent tool for assessing diverse moisture conditions, encompassing regions from unvegetated landscapes to those exhibiting dense vegetative growth (Sandholt, Rasmussen & Andersen, 2002). Its reliability, however, may be affected by dynamic changes in vegetation cover and the intricacies of the terrain, complicating the clear demarcation between arid and humid regions (Shi et al., 2020). On a more expansive scale, the variability introduced by distinct topographical elements translates to differing influences, including precipitation, ambient temperature, and evapotranspiration rates. Such factors play pivotal roles in determining vegetation distribution and heterogeneity in a given area, subsequently affecting its climatic and ecological equilibrium. The innate variability of soil attributes amplifies the uncertainty associated with TVDI’s soil moisture predictions (Maduako et al., 2017). Previous studies posited that the TVDI was best suited for areas exhibiting limited topographical diversity (Rahimzadeh-Bajgiran, Omasa & Shimizu, 2012). However, modern enhancements in the methodology have expanded its applicability to larger expanses, including the Eurasian landmass. Yet, the precise adaptability of TVDI in certain locales warrants further exploration, emphasizing the value of regional customization to bolster prediction precision and diminish potential inaccuracies.

Our study further revealed that an intensifying trend in the severity of drought episodes in Guangdong Province over the years.In recent years, the drought events have been longer in duration, more widespread in impact, and caused more severe economic losses. The spatial characteristics of the distribution of drought events in Guangdong Province also show some changes. Although drought events in Guangdong Province occur throughout the province, there are obvious differences in the frequency and severity of drought events in different regions. The unique geographical positioning of Guangdong Province means its drought patterns are modulated by climatic elements, including monsoons and typhoons. Factors such as typhoon-induced precipitation in summers could modulate drought intensity and reach within the province. The spatial and temporal characteristics of droughts in Guangdong Province are complex, and the combined effects of multiple factors need to be considered. Through an in-depth study of drought monitoring indicators and data, the spatial and temporal characteristics of drought in Guangdong Province can be more accurately grasped.

Conclusions

This study aims to evaluate the effectiveness of two TVDI indices for drought monitoring in Guangdong Province and to examined drought spatiotemporal variations in Guangdong Province during the 20 years from 2000 to 2019. The results show that TVDINDV I responds better to drought in Guangdong Province than TVDIEV I. Meanwhile, the study also identifies the primary drought-prone areas within Guangdong as the Leizhou Peninsula in the south and the central Pearl River Delta region. Additionly, we analyzed the phenomenon of consecutive winter and spring droughts in Guangdong Province within the last 20 years, and the area coverage of each drought grade was 42%∼64.6% for light drought, 6.96%∼27.92% for medium drought, and 0.002%∼1.84% for heavy drought. Among them, the most serious winter-spring drought in the province in 2003, with a drought coverage of 84.2%, and the lowest drought area coverage of 49.02% in 2009. Due to the changes in climatic conditions, the frequency and area of droughts and the duration of long-term droughts have increased sharply. In response to the changing trend of the geographical distribution of drought, the relevant departments should develop corresponding water conservation measures and scientific allocation of water resources to ensure normal socio-economic development.

Supplemental Information

Supplemental Information 1 TVDI code in Guangdong Province

Click here for additional data file.

Supplemental Information 2 Author-Cover-Page-Template

Click here for additional data file.

Additional Information and Declarations

Competing Interests

Author Contributions

Data Availability

The authors declare there are no competing interests.

Ailin Chen conceived and designed the experiments, performed the experiments, analyzed the data, prepared figures and/or tables, authored or reviewed drafts of the article, and approved the final draft.

Jiajun Jiang conceived and designed the experiments, prepared figures and/or tables, and approved the final draft.

Yong Luo performed the experiments, analyzed the data, prepared figures and/or tables, and approved the final draft.

Guoqi Zhang analyzed the data, prepared figures and/or tables, and approved the final draft.

Bin Hu performed the experiments, prepared figures and/or tables, and approved the final draft.

Xiao Wang analyzed the data, authored or reviewed drafts of the article, and approved the final draft.

Shiqi Zhang conceived and designed the experiments, prepared figures and/or tables, authored or reviewed drafts of the article, and approved the final draft.

The following information was supplied regarding data availability:

The TVDI code is available in the Supplemental File.

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
