# Peer review of "Temperature vegetation dryness index (TVDI) for drought monitoring in the Guangdong Province from 2000 to 2019"

_PeerJ, doi:10.7717/peerj.16337_

## Round 0.1 · original submission · Major Revisions

The study overall is interesting but there are some flaws that after major revisions and can be accepted for publication. Currently, a detailed and deep discussion is missing. In addition, the manuscript needs to be polished by a fluent English speaker.

**Language Note:** The Academic Editor has identified that the English language must be improved. PeerJ can provide language editing services - please contact us at copyediting@peerj.com for pricing (be sure to provide your manuscript number and title). Alternatively, you should make your own arrangements to improve the language quality and provide details in your response letter. – PeerJ Staff

Reviewer 1 ·

Basic reporting

The paper is clear. But I didn't see the tables in the submitted paper. And the English needs to be improved.

Experimental design

Research question well defined, relevant & meaningful.
But the method lacks novelty.

Validity of the findings

1. Data source and preprocessing: The paper lacks detailed information on possible data biases or noise, and does not provide enough explanation regarding the selection and acquisition of relevant satellite remote sensing data. The process of choosing preprocessing methods is not well-described.

2. Model selection and evaluation: The paper only discusses one drought monitoring index, the Temperature Vegetation Dryness Index (TVDI), and does not evaluate other remote sensing indices that could be used for drought monitoring, which may limit the effectiveness of the approach. The optimization methods and parameter selection process for TVDI lack a comprehensive description and explanation. Additionally, the evaluation methods may require a more thorough and comprehensive explanation, including the applicability and reliability of the methods.
3. The paper is lack of discussion of the physical mechanisms underlying the relationship between the TVDI and soil moisture or drought conditions. While TVDI is a widely used index in drought monitoring, the underlying physical mechanisms behind the relationship between the LST and VI are not always clear and may vary depending on the region and environment. Additionally, the spatial resolution of the remote sensing data used in this paper may also limit the accuracy and applicability of the TVDI index, as it may not fully capture the heterogeneity of the land surface conditions.

Reviewer 2 ·

Basic reporting

See below

Experimental design

See below

Validity of the findings

See below

Additional comments

Introduction, results and discussion part: Please try to include some related works from a wider geographical spread:

Line 181: please add a part to the section about the spatial resolution of the used satellite images.

Please to check/compare/cite how to make the seasonal/monthly satellite images from similar studies

Please try to add some more information about how to transfer the 8 days images to monthly/seasonal.

Figure 1: the workflow has some typo mistake: MOD11A2 and not MODA11A2, MOD13A3 and not MODA13A3.

figure2: please show all the 25 used gauge station’s points on the study area map.

LINE 186: why you haven’t used the MOD13Q1 instead of the MOD13A3? The first one could be able to have a better resolution that the used images.

Lines 292-294: add a figure showing the processing (before and after LST correction).

Figure 4: What is unit of the Y axis of the histogram? Is it square kilometer? Write it. SAME FOR FIGURE 5.

Part “t”: there is typo mistake. It is October not November. Correct it.

Figure 9 & 10: What is the exact benefit of using a, b , c, …. In the figure? I think it can be deleted.

---

## Round 0.2 · accepted · Accept

I think this manuscript is ready for publication.